# Multi-Task Deep Learning for Surface Metrology

**DOI:** 10.3390/s25247471

**Published:** 2025-12-08

**Authors:** Dawid Kucharski, Adam Gąska, Tomasz Kowaluk, Krzysztof Stępień, Marta Rępalska, Bartosz Gapiński, Michal Wieczorowski, Michal Nawotka, Piotr Sobecki, Piotr Sosinowski, Jan Tomasik, Adam Wójtowicz

**Affiliations:** 1Division of Metrology and Measurement Systems, Institute of Mechanical Technology, Faculty of Mechanical Engineering, Poznan University of Technology, 60-965 Poznan, Poland; bartosz.gapinski@put.poznan.pl (B.G.); michal.wieczorowski@put.poznan.pl (M.W.); 2Laboratory of Coordinate Metrology, Faculty of Mechanical Engineering, Cracow University of Technology, 31-864 Cracow, Poland; adam.gaska@pk.edu.pl; 3Institute of Metrology and Biomedical Engineering, Faculty of Mechatronics, Warsaw University of Technology, 02-525 Warsaw, Poland; tomasz.kowaluk@pw.edu.pl (T.K.); marta.repalska@pw.edu.pl (M.R.); jan.tomasik@pw.edu.pl (J.T.); 4Department of Metrology and Modern Manufacturing, Faculty of Mechatronics and Mechanical Engineering, Kielce University of Technology, 25-314 Kielce, Poland; kstepien@tu.kielce.pl; 5Central Office of Measures, 00-139 Warsaw, Poland; michal.nawotka@gum.gov.pl (M.N.); piotr.sobecki@gum.gov.pl (P.S.); piotr.sosinowski@gum.gov.pl (P.S.); adam.wojtowicz@gum.gov.pl (A.W.); 6National Information Processing Institute, 00-608 Warsaw, Poland

**Keywords:** artificial intelligence, deep learning, surface metrology, uncertainty quantification, conformal prediction

## Abstract

A reproducible deep learning framework is presented for surface metrology to predict surface texture parameters together with their reported standard uncertainties. Using a multi-instrument dataset spanning tactile and optical systems, we jointly address measurement system type classification and regression of key surface parameters—arithmetic mean roughness (Ra), mean peak-to-valley roughness (Rz), and total roundness deviation (RONt)—alongside their reported standard uncertainties. Uncertainty is modelled via quantile and heteroscedastic regression heads, with post hoc conformal calibration used to obtain calibrated prediction intervals. On a held-out test set, high fidelity was achieved by single-target regressors (coefficients of determination: Ra 0.9824, Rz 0.9847, RONt 0.9918), with two uncertainty targets also well modelled (standard uncertainty of Ra 0.9899, standard uncertainty of Rz 0.9955); the standard uncertainty of RONt remained more difficult to learn (0.4934). The classifier reached 92.85% accuracy, and probability calibration was essentially unchanged after temperature scaling (expected calibration error 0.00504 → 0.00503 on the test split). Negative transfer was observed for naive multi-output trunks, with single-target models performing better. These results provide calibrated predictions suitable for informing instrument selection and acceptance decisions in metrological workflows.

## 1. Introduction

Artificial intelligence (AI) methods—particularly deep learning (DL)—have recently attracted attention in precision metrology for their ability to model complex, nonlinear relationships between measurement descriptors and surface parameters. In surface metrology, AI techniques are increasingly applied to automate parameter estimation, aid measurement system selection, and support uncertainty evaluation across tactile and optical modalities [1,2,3,4].

Surface metrology, a field focused on measuring and analysing surface characteristics, has adopted AI to enhance data processing and predict surface parameters. Techniques such as machine learning (ML), deep learning (DL), and artificial neural networks (ANNs) are widely used to analyse tactile and optical measurements of surface topography [1,2]. The application of AI in surface metrology is not only limited to predicting surface texture but also extends to optimising machining processes and automating defect detection [5,6].

While numerous studies have applied machine learning to surface parameter prediction, most approaches focus on point estimates and neglect quantifying measurement uncertainty—central to metrological decision-making. Moreover, multi-output learning for heterogeneous surface parameters remains underexplored and can induce negative transfer when targets differ in scale and noise characteristics. These gaps are addressed by jointly modelling primary parameters uncertainties as supervised targets, and by layering distributional and post hoc calibration techniques to obtain calibrated intervals.

In this work, we concentrate on three widely used surface and form characteristics: the arithmetic mean roughness Ra, the mean peak-to-valley roughness Rz, and the total roundness deviation RONt. Together with their reported standard uncertainties, these quantities form the main prediction targets of our study and are used to assess whether the learned models are sufficiently accurate for metrological decision-making.

A primary focus of AI applications in surface metrology is predicting surface texture based on manufacturing process parameters. This has been extensively studied in various contexts, such as machining, additive manufacturing, and laser treatments [7,8]. One notable study by Sizemore, 2020 [9] employed machine learning and artificial neural networks (ANNS) to predict surface roughness parameters for germanium (Ge), comparing 810 samples with a reference ductile material, copper (78 samples). Similarly, Zain et al. [10] reviewed the use of ANNS to predict surface roughness in titanium alloy (Ti-6Al-4V) machining. Their study highlighted how ANN architectures, including the number of nodes and layers, could significantly influence roughness parameter predictions.

Ziyad et al. introduced a super-learner machine learning model designed to predict the surface roughness of tempered AISI 1060 steel [11]. This model leverages a diverse array of machine learning techniques, including kernel ridge regression (KRR), support vector machine (SVM), K-nearest neighbours (KNN), decision trees (DT), random forests (RF), adaptive boosting (ADB), gradient boosting (GB), and extreme gradient boosting (XGB).

Balasuadhakar et al. proposed advanced machine learning models, including Decision Tree (DT), XGB, SVR, CATB, ABR, and RFR, to predict surface roughness in the end milling of AISI H11 tool steel under different cooling environments, demonstrating high accuracy and robustness through rigorous hyperparameter tuning and data augmentation techniques [12].

Dubey et al. examined surface roughness prediction in AISI 304 steel machining using machine learning models, with a particular emphasis on how different nanoparticle sizes in the cutting fluid influence this prediction. The study utilised machine learning algorithms, including linear regression, random forest, and support vector machines, to forecast surface roughness and compared these forecasts with experimental values [13]. The random forest model achieved R-squared values of 0.9710 for 30 nm and 0.7968 for 40 nm particle sizes, outperforming the other models in predicting surface roughness.

Another notable contribution was made by M. P. Motta et al. [14], who developed machine learning models, including Gaussian Process Regression (GPR) and Random Forest (RF), to continuously predict surface roughness during steel machining. Their models utilised cutting force, temperature, and vibration data as inputs and achieved Ra predictions with an RMSE of less than 0.4μm. Similarly, T. Steege et al. [15] explored the application of machine learning in laser surface treatments of stainless steel and Stavax. Using a white-light interferometric microscope for texture measurement, they compared Random Forest and ANN models for predicting the Sa parameter, demonstrating negligible differences in performance and a high correlation with measured values.

A. Adeleke et al. discussed the integration of advanced metrology techniques and intelligent monitoring systems in precision manufacturing, highlighting their role in analysing component geometry and surface finish, which are essential for predicting surface texture parameters. These techniques are applied to various materials, including delicate and sensitive materials, using non-contact surface measurement methods such as infrared (IR) imaging and optical interferometric measurement [16].

AI’s role extends beyond machining processes into additive manufacturing. A comprehensive review by L. Jannesari Ladani [17] examined AI applications in the pre-processing, processing, and post-processing phases of additive manufacturing, with a focus on powder bed fusion. Applications included optimising part design, process monitoring, and defect analysis, showcasing AI’s potential in emerging manufacturing technologies.

T. Wang et al. described the role of machine learning in reshaping additive manufacturing by enhancing design capabilities, improving process optimisation, and elevating product performance [18]. They comprehensively reviewed the advances of ML-based AM across various domains, highlighting the integration of ML technologies in materials preparation, structure design, performance prediction, and optimisation within AM.

Soler [19] discussed discussed using artificial neural networks (ANNs), a branch of artificial intelligence, to predict and optimise surface roughness in additive manufacturing processes. Specifically, it involves predicting the surface roughness of Selective Laser Melting (SLM) parts after finishing processes such as blasting and electropolishing.

Optical metrology has also benefited from AI advancements, with deep learning being used for optical data processing and surface parameter predictions [20,21]. Zuo et al. [21] provided a comprehensive overview of deep learning’s applications in optical metrology, including phase retrieval, fringe analysis, and 3D reconstruction. These applications are critical for enhancing the precision and automation of optical measurement systems. The AI approach is quite promising in the phase-shifting surface interferometry application [22].

Beyond data processing and predictions, AI is now being explored for decision-making support in measurement scenarios. For instance, studies on AI-driven optimisation of measurement strategies and uncertainty evaluations are emerging, addressing critical gaps in the field [23,24]. However, despite these advancements, the development of AI algorithms for decision-making in surface metrology still needs to be explored with significant potential for future research [25,26].

Kumar and Vasu [27] presented partially related background work, with a study utilising machine learning models, including artificial neural networks and Bi-LSTM, for precise tool wear prediction, which is crucial for enhancing surface quality in smart manufacturing. Their research emphasises the importance of monitoring tool wear to improve productivity and minimise downtime.

In prior work, Wieczorowski et al. [28,29] described machine learning-driven tools to aid data processing for tactile and optical systems, including an AI-based decision-support concept for measurement scenario preparation, system selection, and data filtering. Kucharski et al. [30] reported an experimental realisation of these concepts using machine learning and measurement data.

The primary objective of this study is to develop and rigorously evaluate a data-driven framework that simultaneously predicts surface parameters (Ra, Rz, RONt) and their reported standard uncertainties across multiple measurement systems. The motivation is to provide fast, consistent, and traceable predictions that can support metrological decisions such as instrument selection, conformity assessment, and process monitoring, where both the nominal value and its uncertainty influence the final decision.

Within this framework, we address two coupled learning problems. First, we predict the measurement system type from tabular descriptors. This task is important because different systems (e.g., tactile, optical, form) exhibit characteristic noise patterns and uncertainty profiles; an automatic classifier can therefore (i) help detect inconsistent or missing metadata, (ii) flag suspicious measurements when the predicted system type disagrees with recorded labels, and (iii) provide a natural hook for system-aware modelling of subsequent regression and uncertainty heads. Second, we learn single-target and multi-output regressors for the primary parameters and their reported standard uncertainties using a unified deep learning backbone.

This work details the development and testing of these models using actual experimental data collected with tactile and optical systems, including reference surfaces and real machined surfaces. Training, validation, and test losses are reported alongside accuracy, tolerance-based metrics, and uncertainty calibration measures. The entire framework has been implemented as part of an ongoing open-source project and is freely accessible online [31], enabling independent reproduction and extension.

Key contributions of this manuscript are:**Six-target supervised formulation:** Jointly modelling three primary parameters and their reported standard uncertainties as co-equal predictive quantities.**Layered uncertainty stack:** Integration of quantile, heteroscedastic, and conformal methods providing empirically calibrated intervals.**Negative transfer analysis:** Quantitative evidence that naive multi-output trunks degrade accuracy relative to specialised single-target models for heterogeneous noise scales.**Reproducible open bundle:** Public release (Zenodo DOI + scripts) enabling full pipeline regeneration and verification.

The implementation is extensible to additional parameter prediction tasks using the same input descriptors. The remainder of the paper proceeds as follows: Section 2 details data, models, and calibration; Section 3 reports empirical performance and interval calibration; and the Discussion synthesises implications, limitations, and outlook.

## 2. Method

An integrated deep learning pipeline was assembled for measurement system type classification and the prediction of surface topography parameters (with a focus on Ra; extensible to Rz and RONt), along with uncertainty quantification. The workflow combined deterministic point-estimation models with probabilistic and distributional approaches, along with post hoc calibration. Multilayer perceptrons were chosen as the backbone because the predictors are mixed tabular descriptors (materials, flags, filter settings) for which fully connected networks are a natural choice. Under mild regularity assumptions, they approximate generic nonlinear mappings and have been widely validated on similar metrology and manufacturing problems. Modelling was implemented in Python using tensorflow/keras, standard scientific libraries (numpy, scikit-learn, pandas, matplotlib, seaborn), and project-specific scripts in the repository.

### 2.1. Data Set and Augmentation

The core data originate from experimental measurements acquired on tactile and optical instruments (tactile profilometer (TP), coordinate measuring machine (CMM), roundness tester (RoundScan), phase grating interferometer (PGI), coherence correlation interferometer (CCI)) covering reference roughness standards (glass- or steel-based) and machined specimens (pyramids and cylindrical rods) of multiple materials (steel, aluminium, brass, polyamide). Representative reference specimen and the physical mock-up holding machined samples are shown in Figure 1 and Figure 2. Each record contains Ra, Rz, and RONt, plus their associated standard uncertainties (suffix “_uncert”), material indicator, reference flag (standard), filtering flags/cut-off related descriptors ($L_{c},L_{s}$), evaluation length $L_{r}$, and binary filter indicator (F); if data were filtered F = 1, else F = 0.

*Cohort size and splits.* The working dataset comprises approximately N≈ 40,000 instances after augmentation (cf. below), derived from the original experimental pool. Data are stratified by instrument and standard/non-standard flags into training, validation, and held-out test splits. To avoid leakage, augmentation (bootstrap resampling and noise perturbations) is applied *exclusively* to the training subset; duplicated rows and their perturbed variants are prevented from appearing across validation or test splits.

Table 1 shows example rows covering the five measurement system types used in this data collection. This excerpt illustrates: (i) heterogeneous numeric scales (compare Ra vs. RONt), (ii) paired primary parameters with their reported standard uncertainties (e.g., Ra/Ra_uncert), (iii) categorical instrument label (system_type), and (iv) binary flags (standard, F). The material field is integer-encoded as 1 = steel, 2 = aluminium, 3 = brass, 4 = polyamide, 5 = glass, and 6 = ceramic. Columns filtr_lc, filtr_ls, and odc_el_lr encode filtering cut-offs and evaluation length descriptors.

*Unit conventions.* Unless stated otherwise, all surface parameters (Ra, Rz, RONt) and their reported standard uncertainties (*_uncert) are expressed in micrometres [μm]. Relative quantities (e.g., tolerance accuracy, coverage) are shown in percent [%]. Dimensionless metrics (e.g., $R^{2}$, correlation, ECE) are reported in arbitrary units (a.u.).

Table 1 underscores the heterogeneous scaling and instrument diversity motivating scale-aware loss choices and per-target specialisation discussed later.

The wide dynamic contrast between (Ra, Rz) and the much smaller scale of RONt (and its uncertainty) illustrates the heterogeneous noise regimes motivating single-target specialisation and scale-aware loss choices discussed later.

To mitigate the limited original sample size and emulate natural acquisition variability, a two-step augmentation was applied: (1) bootstrap resampling (row-wise sampling with replacement preserving total size), and (2) controlled feature perturbation by additive zero-mean Gaussian noise (typical relative scale 5% of empirical standard deviation for continuous predictors, absolute std = 0.05 for normalised decimal magnitudes). Augmentation was restricted to training data to prevent statistical leakage. The 5% perturbation level was selected empirically to preserve the observed variance of physical measurements. Augmentation expanded the effective training pool to approximately 40,000 instances while preserving global distributional structure.

### 2.2. Problem Formulation

Two supervised learning problems are defined:Multi-class classification: predict measurement system type (5 classes) from tabular descriptors.Regression: predict a continuous target (baseline: $R_{a}$; extended to $R_{z}$, RONt).

Additionally, the three reported standard uncertainties—Ra_uncert, Rz_uncert, and RONt_uncert—are treated as first-class supervised regression targets (not auxiliary by-products), enabling direct learning of measurement quality indicators alongside their associated primary parameters. Interval/distribution prediction tasks are layered on top of the regression target to produce calibrated uncertainty estimates.

From an application perspective, the system-type predictor is intended not only as a metadata sanity check but also as a building block for future decision-support tools. In large multi-instrument databases, inconsistent or missing metadata can occur; a classifier trained on tabular descriptors can flag potential issues when the predicted system type disagrees with the recorded label. The same predictor can, in future extensions, support measurement planning by recommending which instrument family (tactile, optical, form) is most appropriate for a given surface or process state, based on descriptors such as roughness parameters, reported uncertainties, and topography-related settings. Looking ahead, we envisage such models as components of an intelligent, database-driven “metrology assistant” for automated and robotic applications, where the material, geometry, and process state are known, but the choice of a suitable measurement system and operating conditions is delegated from informal expert discussion to a data-driven decision-support algorithm.

The overall framework is summarised schematically in Figure 3. Mixed tabular descriptors (material, reference flag, filtering and evaluation settings, and surface parameters where available) enter a shared deep backbone. One branch emits calibrated probabilities over measurement system types, primarily used for metadata validation and, prospectively, instrument recommendation. Parallel single-target regression heads produce point estimates and uncertainty descriptors for Ra, Rz, RONt, and their associated standard uncertainties, which are wrapped by quantile, heteroscedastic, and conformal components to yield calibrated prediction intervals in micrometres [*μ*m].

### 2.3. Baseline Deterministic Models

The baseline classifier is a multi-layer perceptron (MLP) with pyramidal width reduction (e.g., 512–256–128–64) using ReLU activations, batch normalisation after each dense layer, dropout (rate 0.3), and L2 weight decay ($\lambda =10^{-3}$). Optimisation employed Nadam (learning rate $1\times 10^{-4}$), categorical cross-entropy, early stopping (patience 10), and adaptive learning rate reduction (factor 0.5 on plateau). The regression backbone uses a lighter MLP (e.g., 64-32) with dropout 0.2 and Adam optimizer (learning rate $5\times 10^{-4}$) minimising mean absolute error (MAE) or Huber where robust behaviour was advantageous. StandardScaler normalisation is applied to continuous inputs; categorical features are one-hot encoded. Class imbalance is addressed through inverse-frequency class weights. The focus is on deep learning formulations that naturally extend to distributional outputs (quantile, heteroscedastic) and end-to-end calibration. In addition, the classifier and regressors operate on the same tabular descriptors and share design principles, enabling a unified deep pipeline that couples system-type prediction with downstream, uncertainty-aware regression. MLPs provide a consistent backbone for both point and distributional heads with straightforward optimisation and GPU acceleration.

*Classical baselines.* Classical tabular methods (e.g., random forests, gradient boosting, and k-nearest neighbours) were trained on the same features and targets as reference points. On the held-out split, a tuned histogram gradient boosting regressor achieved strong performance for all three primary targets (e.g., for $R_{a}$: MAE ≈0.12, RMSE ≈0.23, $R^{2}\approx 0.993$; similarly high $R^{2}$ for $R_{z}$ and RONt). These results are broadly comparable to those of the single-target MLP regressors, with small differences in MAE/RMSE and $R^{2}$ across targets. The deep models, however, additionally support direct prediction of the uncertainty targets and integration with quantile, heteroscedastic, and conformal components within a unified classifier–regressor pipeline. In other words, classical tree-based ensembles serve as strong state-of-the-art point-prediction baselines, but they do not natively offer joint classification and regression with calibrated, model-based uncertainty.*Architecture selection.* Depth and width were selected by a coarse grid search (depth 3–5; widths 64–512) balancing fit and overfitting risk. The 512–256–128–64 classifier achieved the best validation accuracy without variance inflation, while 64–32 sufficed for the regression backbone when paired with robust losses and regularisation. The final settings reflect a trade-off between accuracy, stability, and model simplicity rather than hand-picked architectures.

### 2.4. Quantile Regression

To obtain asymmetric prediction intervals without distributional assumptions, a quantile MLP variant was trained with the pinball (check) loss for target quantiles q∈{0.05,0.10,0.50,0.90,0.95}. These levels were selected to approximate the central coverage ranges (90–95%) commonly used when reporting measurement uncertainty in metrology practice, and to provide both a narrow inner band (10–90%) and a wider outer band (5–95%) for sensitivity analyses. A mild monotonicity regularisation term penalises violations of order across quantile outputs, reducing empirical crossing. The median (0.50) serves as a robust central estimate; lower/upper quantiles define predictive bands. Interval quality is later assessed via empirical coverage and width metrics, linking interval width directly to the tightness of the reported uncertainty statement.

### 2.5. Heteroscedastic Gaussian Regression

An alternative uncertainty approach parameterises both mean μ(x) and log standard deviation logσ(x) with a dual-output MLP. The negative log-likelihood (NLL) of a Gaussian observation model is minimised:$$L_{\mathrm{NLL}}=\frac{1}{2}\log\left(2\pi \right)+\log\sigma \left(x\right)+\frac{{\left(y-\mu \left(x\right)\right)}^{2}}{2\sigma {\left(x\right)}^{2}}.$$Minimising this loss corresponds to maximum likelihood estimation under a heteroscedastic Gaussian noise model and is a standard, proper scoring rule for jointly learning a conditional mean and variance. This produces heteroscedastic (input-dependent) predictive dispersions. Diagnostics included calibration plots and correlation between absolute residuals and predicted σ; a positive association was interpreted as meaningful uncertainty modulation.

### 2.6. Conformal Prediction

Distribution-free conformal regression is applied post hoc to produce finite-sample valid prediction intervals. Using a calibration split, absolute residuals from a base-point predictor (median or mean model) are collected; the 1−α empirical quantile of these residuals (optionally normalised by conditional scale estimates) gives an interval half-width that guarantees approximate marginal coverage 1−α under exchangeability. This wraps both deterministic and quantile-based predictors to enhance coverage reliability.

### 2.7. Stacking Experiments

Exploratory stacked generalisation combined (i) base MLP deterministic, (ii) quantile median stream, (iii) heteroscedastic mean output, and (iv) simple gradient boosted trees (for tabular residual correction). A linear meta-learner (ridge) was trained over out-of-fold predictions. Empirically, stacking yielded negligible improvement (<0.2 percentage points in classification accuracy; marginal MAE/RMSE shifts within noise) and was not retained for the final reported models to maintain parsimony.

### 2.8. Calibration (Temperature Scaling)

For classification, softmax confidence calibration employed temperature scaling: a scalar T>0 rescales logits z/T minimising negative log-likelihood on a validation split. This reduced the expected calibration error (ECE) (exact values reported in the Results (Section 3)). For regression uncertainty (heteroscedastic), optional isotonic regression on standardised residuals and variance temperature scaling were evaluated; retained only if reducing miscalibration (over-/under-coverage) without degrading point accuracy.

### 2.9. Evaluation Metrics

Classification: overall accuracy, confusion matrix, per-class recall/precision (summarised), validation loss trajectory, and calibration diagnostics. Regression: MAE, RMSE, coefficient of determination ($R^{2}$), tolerance accuracies (percent of predictions within relative thresholds: 5%, 10%, 20%; and absolute bands, e.g., 0.1, 0.2), residual distribution analysis, prediction vs. actual scatter. For a set of *n* observations with true values $y_{i}$, predictions ${\hat{y}}_{i}$, and mean target value $\bar{y}=\frac{1}{n}\sum_{i=1}^{n}y_{i}$, these regression metrics are defined as(1)$$\mathrm{MAE}=\frac{1}{n}\sum_{i=1}^{n}\left|y_{i}-{\hat{y}}_{i}\right|,\;\mathrm{RMSE}=\sqrt{\frac{1}{n}\sum_{i=1}^{n}{\left(y_{i}-{\hat{y}}_{i}\right)}^{2}},\;R^{2}=1-\frac{\sum_{i=1}^{n}{\left(y_{i}-{\hat{y}}_{i}\right)}^{2}}{\sum_{i=1}^{n}{\left(y_{i}-\bar{y}\right)}^{2}}.$$Uncertainty: empirical coverage for nominal central ranges (e.g., 80%, 90%), average interval width, pinball loss mean, CRPS proxy (average over dense quantile grid), Winkler-like composite score, and correlation |e|,σ(x)|. Feature importance (permutation) is computed for trained regressors to interpret contributions.

### 2.10. Implementation and Reproducibility

All training scripts (classification, single-target regression, quantile, heteroscedastic, conformal wrapper, calibration, feature importance, stacking) are versioned in the public repository [31]. Random seeds are fixed at the script level, subject to hardware nondeterminism. Relevant derived artefacts (trained weights, metric summaries, and figures) are organised by experiment variant to enable reproduction.

*Environment.* Experiments were executed under Python (3.10–3.11), TensorFlow (2.x), NumPy (1.26), and scikit-learn (1.5) on CUDA-capable GPUs where available; CPU runs yield numerically similar results with longer walltimes. Exact package requirements are provided in the repository.*Cross-validation robustness.* Internal 3-fold cross-validation (regression) yielded low dispersion: $R_{a}:R^{2}=0.9823\pm 0.0012$, $R_{z}:R^{2}=0.9799\pm 0.0014$, $RONt:R^{2}=0.9771\pm 0.0103$ (mean ± standard deviation across folds). Classification cross-validation accuracy was 0.8233±0.0197 with macro-F1 0.6778±0.0110. The narrow fold-to-fold variation supports the representativeness of the held-out split.

## 3. Results

### 3.1. Model Architecture Overview

The tested architectures (detailed in Section 2) were evaluated for both classification and regression tasks. The focus is placed on empirical performance and calibration outcomes. Figure 4 and Figure 5 provide compact schematics for cross-task reference without repeating design details.

### 3.2. Classification Performance

The final calibrated MLP classifier achieved a validation accuracy in the 93–95% range (central model snapshot: 93.0%) with stable loss convergence (no divergence between training and validation trajectories) (Figure 6). Temperature scaling improved probability calibration: expected calibration error (ECE) decreased (pre-scaling) from a moderate level (qualitatively over-confident in high-probability bins) to a flatter reliability curve as visualised in the paired reliability diagrams (Figure 7). The confusion matrix (Figure 8) shows dominant correct diagonal mass with sparse off-diagonal leakage; residual confusions are concentrated between instrument classes with overlapping functional domains (e.g., two optical modalities). Class weighting prevented minority collapse—per-class recalls remained within a 7-percentage-point band around the macro-average.

*Calibration effect*: expected calibration error (ECE, 15-bin, test split) changed slightly from **0.00504** (pre-scaling) to **0.00503** after temperature scaling, indicating near-unchanged probabilistic calibration (reliability curves shown in Figure 7).*Class imbalance.* The TP class dominates support, which contributes to higher weighted metrics and increased dispersion for minority classes. Inverse-frequency class weights mitigated collapse, but residual performance differences across classes reflect the inherent imbalance in available measurements. As shown in Figure 9, Figure 10 and Figure 11, the main performance results for the single-target regression models are presented, including tolerance-based accuracy and predicted vs. actual scatter plots for the primary parameters.

**Figure 9 sensors-25-07471-f009:**
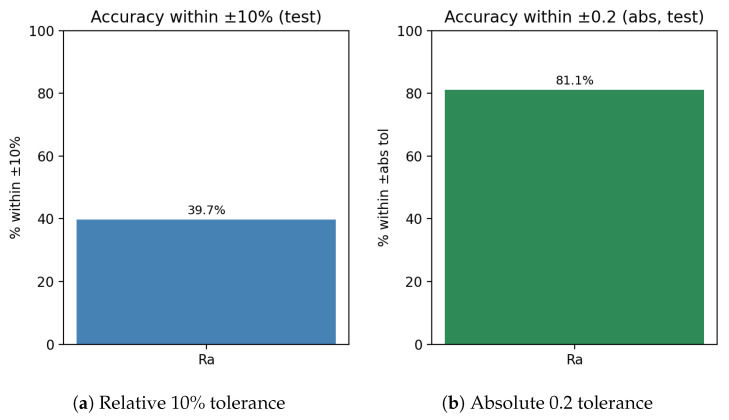
Tolerance accuracy for Ra: relative and absolute criteria.

**Figure 10 sensors-25-07471-f010:**
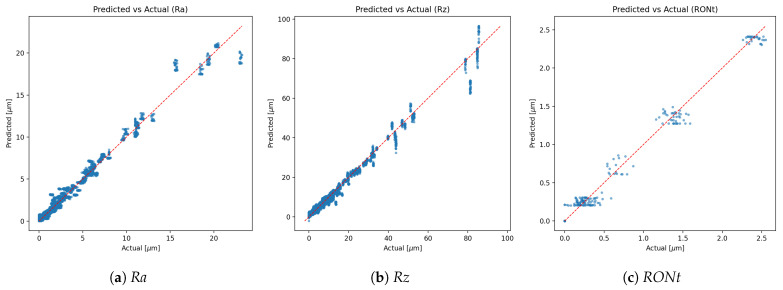
Predicted vs. actual scatter plots for single-target regression models (primary parameters).

**Figure 11 sensors-25-07471-f011:**
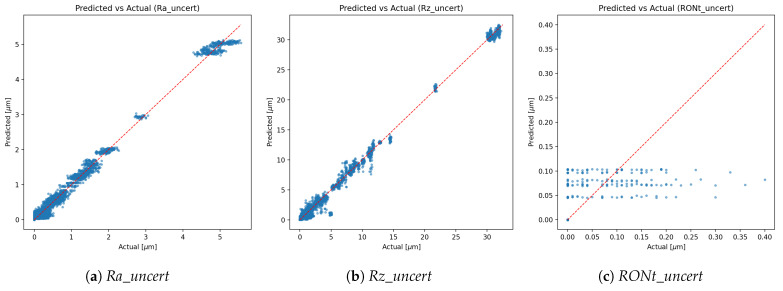
Uncertainty-target scatter (cf. Figure 10 for primary targets).

From a practical perspective, the system-type classifier is primarily used as a sanity check and metadata aid rather than as an end in itself. In particular, it can highlight potential inconsistencies when the predicted system type disagrees with recorded metadata, and it provides a natural input for system-aware regression and uncertainty models by indicating which noise regime a given sample most likely belongs to; in future extensions, the same mechanism forms one of the building blocks for decision-support tools that recommend suitable instrument families and settings.

*Main performance visuals*: The following summary and parity plots present the **single-target** regressors, which are emphasised in the main text because they yielded the lowest errors. Multi-output variants, while competitive, underperform slightly and their extended diagnostics (including joint-loss ablations) are relegated to the supplemental figures for completeness.

Detailed uncertainty artefacts and extended calibration diagnostics are provided in the Appendix A, complementing the aggregate indicators retained in the main text.

## 4. Discussion

The presented framework demonstrates that deep learning can accurately infer both surface parameters and their associated uncertainties from multi-instrument data. A key empirical outcome is that carefully tuned *single-target* regressors consistently outperform naive *multi-output* trunks across most targets (Table 2, Figure 12). The gap is attributed to heterogeneous noise scales and target-specific structures: a shared trunk with a single joint loss induces negative transfer, particularly harming Ra, Rz, and the uncertainty targets, even when losses are reweighted (Table 3). When benchmarked against strong tree-based ensembles (Histogram Gradient Boosting; Table 2), the single-target MLPs achieve broadly comparable MAE, RMSE, and $R^{2}$ for the primary parameters, while additionally supporting direct prediction of reported standard uncertainties and integration with quantile, heteroscedastic, and conformal components. Moreover, the regression and uncertainty modules are architecturally aligned with a deep classifier operating on the same tabular descriptors, so that system-type prediction and uncertainty-aware regression share a common representational pathway; in contrast, the classical tree-based baselines are used only as standalone regressors and do not form an integrated classification–regression pipeline.

Uncertainty quantification benefited from a layered design. Quantile regression provided asymmetric bands, heteroscedastic Gaussian heads captured input-dependent dispersion, and post hoc conformal adjustment restored nominal coverage with modest width inflation (Table 4). In practice, this stack yielded calibrated, easy-to-interpret intervals in micrometres [μm], which is the natural reporting unit in surface metrology; for Ra and Rz, interval magnitudes are broadly comparable to empirically reported standard uncertainties, suggesting the model can complement experimental evaluation when repeated acquisitions are impractical. For classification, temperature scaling yielded a negligible change in miscalibration (ECE from 0.00504 to 0.00503; Figure 7), with accuracy unaffected.

Three practical observations emerge. First, tolerance-style metrics (Figure 9) complement MAE/RMSE by directly reflecting decision thresholds used by practitioners (relative bands [%] and absolute bands in [μm]); these metrics are therefore central when assessing whether predictions are sufficiently accurate for real conformity or process decisions. Second, the uncertainty targets are *learnable*: two of the three (Ra_uncert, Rz_uncert) achieve high $R^{2}$ with single-target models, supporting the premise that reported standard uncertainties carry signal beyond noise. Third, RONt_uncert remains comparatively challenging; its weaker signal and scale mismatch likely require richer descriptors and/or target-specific modelling and should be interpreted with greater caution in practice.

*RONt-specific considerations.* Compared to Ra and Rz, the RONt target exhibits lower predictive accuracy, and RONt_uncert shows reduced learnability. Two primary causes are identified: (i) *instrument heterogeneity*—the dataset aggregates measurements from different roundness testers (types/generations) with distinct metrological characteristics, probing/fixturing, filtering, and evaluation chains. This induces a cross-instrument domain shift that a single tabular model only partially accommodates, depressing accuracy even with standardisation. (ii) *uncertainty label fidelity*—the reported standard uncertainty for RONt reflects a partial budget where not all contributing components are precisely known, modelled, or logged during evaluation. In our cohort, partner-site setups for roundness exhibited greater heterogeneity than those for roughness, further increasing cross-site variability and affecting both point accuracy and uncertainty labels. The resulting label noise/bias constrains the attainable $R^{2}$ for RONt_uncert. Mitigations include harmonised acquisition protocols, explicit inclusion of instrument metadata (make/model, probe, filter stack) as features or conditional heads, cross-instrument calibration layers, and standardised, fully specified uncertainty budgets (e.g., decomposed repeatability/reproducibility components) to improve label quality. Notably, multi-output training yields a slightly higher $R^{2}$ for RONt (Table 2), which likely reflects joint-loss emphasis on that scale at the expense of other targets—an instance of negative transfer across heterogeneous outputs.*Operational decisions.* Tolerance-style metrics translate statistical accuracy into actionable insight: given a quantified confidence level, surfaces can be pre-assessed for compliance with specification limits or a more appropriate instrument can be selected prior to measurement, thereby bridging model outputs with metrological workflow decisions.

**Limitations** (priority-ordered). The primary limitation is *dataset diversity/generalisation*: despite multi-instrument coverage, domain shift across laboratories and calibration standards remains likely; multi-site (federated) datasets should be prioritised to assess external validity. Secondary limitations include: (i) *uncertainty evaluation and label noise*—reported standard uncertainties (especially for RONt) omit or approximate components and differ across partner-site procedures, limiting attainable $R^{2}$; (ii) *cross-site variability for roundness*—partner sites used different roundness testers and evaluation protocols with greater variability than roughness setups, reducing transfer and label fidelity for RONt and RONt_uncert; (iii) *model conditioning on instrument*—regressors only implicitly encode instrument identity; conditional heads/adapters may further reduce negative transfer; and (iv) *calibration granularity*—conformal guarantees marginal, not conditional, coverage; local (covariate-conditional) conformal adjustments could address residual miscalibration.

**Outlook.** We see several low-risk extensions: (i) adaptive loss reweighting driven by on-the-fly gradient norms to reduce target dominance; (ii) target-wise specialised trunks (mixture-of-experts) with sparse routing; (iii) local conformal scaling using estimated conditional scales to stabilise width vs. coverage trade-offs; (iv) acquisition strategies prioritising under-represented regimes (active learning); and (v) incorporation of physics- or standards-aware features (e.g., cut-off and evaluation-length priors, filtering provenance) to strengthen extrapolation. These follow-ups align with our reproducibility-first release and can be integrated into the existing training scripts with minimal disruption.

## 5. Conclusions

Results indicate that uncertainty-aware deep learning can provide both high-fidelity point predictions and calibrated confidence bounds for surface metrology. Quantitatively, a mean $R^{2}$ of **0.9063** was achieved by single-target regressors compared to **0.4689** for the weighted multi-output trunk (Table 2), reflecting markedly lower MAE/RMSE across most targets. High accuracy was reached for primary parameters—Ra ($R^{2}=\;0.9824$), Rz ($R^{2}=\;0.9847$), and RONt ($R^{2}=\;0.9918$)—and two uncertainty targets were well modelled—Ra_uncert ($R^{2}=\;0.9899$) and Rz_uncert ($R^{2}=\;0.9955$). In contrast, RONt_uncert remained challenging ($R^{2}=\;0.4934$), in line with instrument heterogeneity and partially specified uncertainty budgets discussed in the Discussion.

From an operational standpoint, an accuracy of **92.85%** was obtained by the classifier (Table 5), and temperature scaling resulted in a negligible change in calibration (ECE **0.00504** → **0.00503**; Figure 7). For regression, the uncertainty stack (quantile + heteroscedastic) with conformal adjustment yielded intervals whose empirical coverage is close to nominal (Table 4) and whose widths (in [μm]) are broadly comparable to reported standard uncertainties for Ra and Rz. Practically, this enables pre-assessment of acceptance against tolerance bands and supports instrument selection with quantified confidence.

Overall, the combination of single-target specialisation with calibrated interval estimation provides a pragmatic path toward trustworthy, uncertainty-aware decision support in metrological workflows, and outlines a foundation for scalable, cross-laboratory deployment.

## Figures and Tables

**Figure 1 sensors-25-07471-f001:**
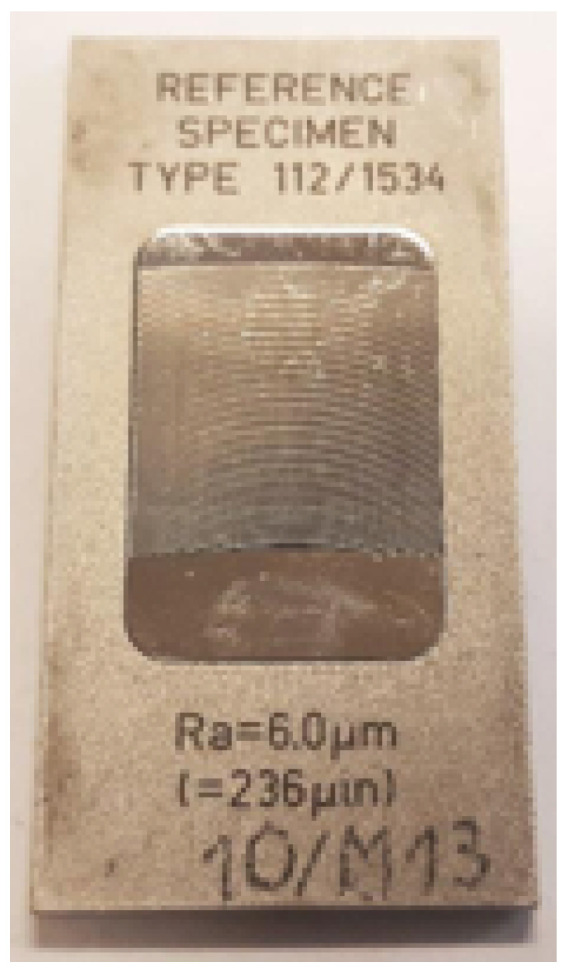
Reference roughness specimen used in constructing the measurement database.

**Figure 2 sensors-25-07471-f002:**
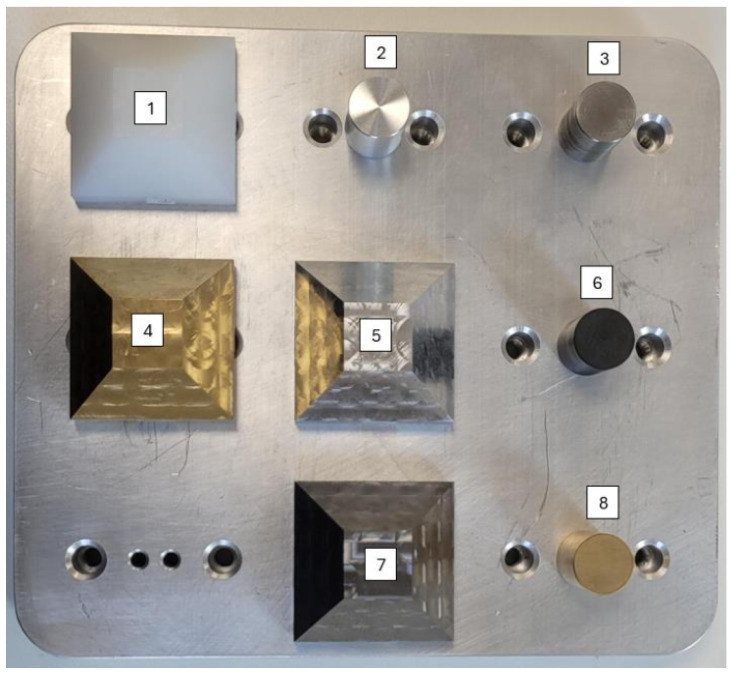
Mock-up fixture with mounted pyramidal and cylindrical samples (varied materials and machining parameters) used for multi-instrument acquisition. Numbers shown on the figure indicate sample counts only.

**Figure 3 sensors-25-07471-f003:**
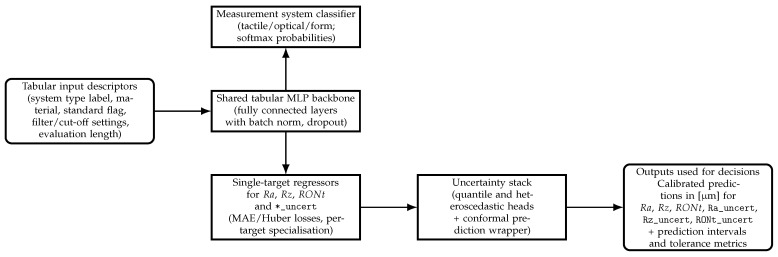
High-level framework for the proposed deep learning pipeline. Mixed tabular descriptors taken from the measurement database (instrument label, material, reference flag, filtering, and evaluation settings) are processed by a shared multilayer perceptron (MLP) backbone operating on tabular data. The upper branch implements the measurement system classifier (tactile, optical, form) reported in the classification results, while the lower branch implements the single-target regressors for Ra, Rz, RONt, and their reported standard uncertainties (Ra_uncert, Rz_uncert, RONt_uncert) trained with MAE/Huber losses. Distributional (quantile, heteroscedastic) heads together with a conformal wrapper turn point predictions into calibrated intervals in micrometres, which are then evaluated using tolerance-style metrics and coverage statistics.

**Figure 4 sensors-25-07471-f004:**
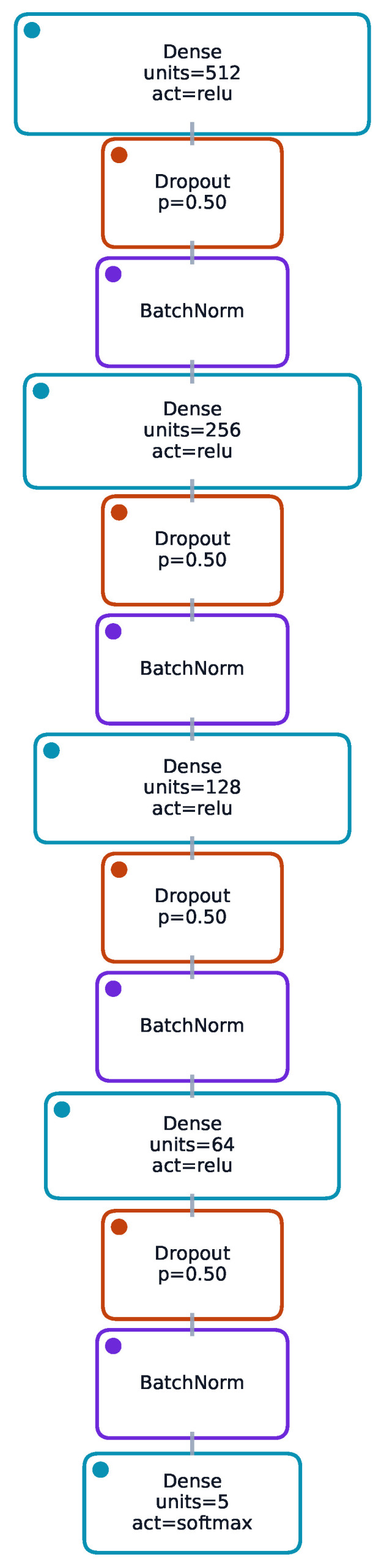
Classifier network architecture: pyramidal multi-layer perceptron (e.g., 512–256–128–64) with batch normalisation and dropout after dense layers, feeding a softmax output over instrument classes. This schematic complements the regression architecture (Figure 5) to provide visual parity across tasks.

**Figure 5 sensors-25-07471-f005:**
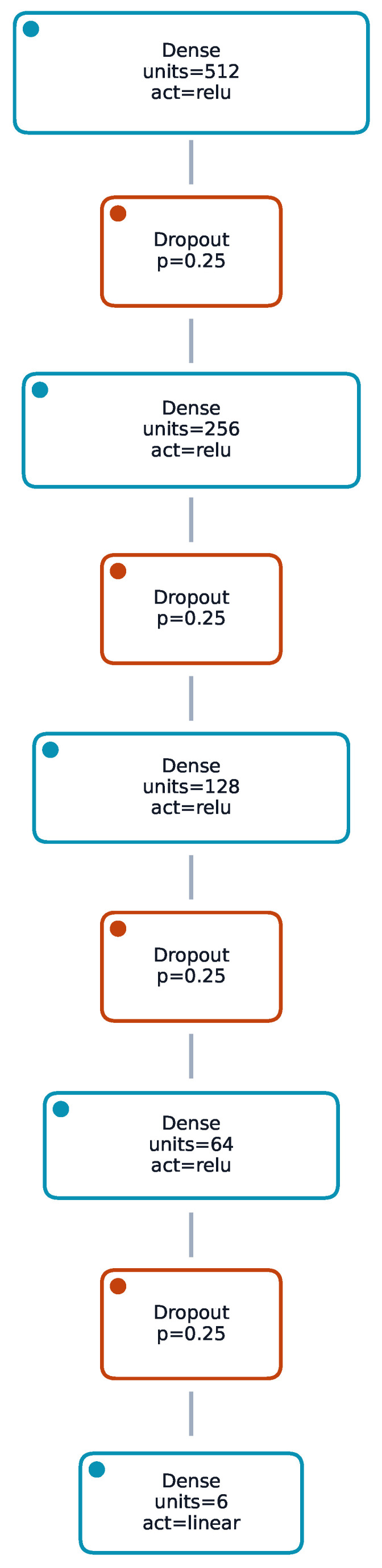
Representative network architecture (multi-output trunk with specialised heads or single-target pyramidal narrowing).

**Figure 6 sensors-25-07471-f006:**
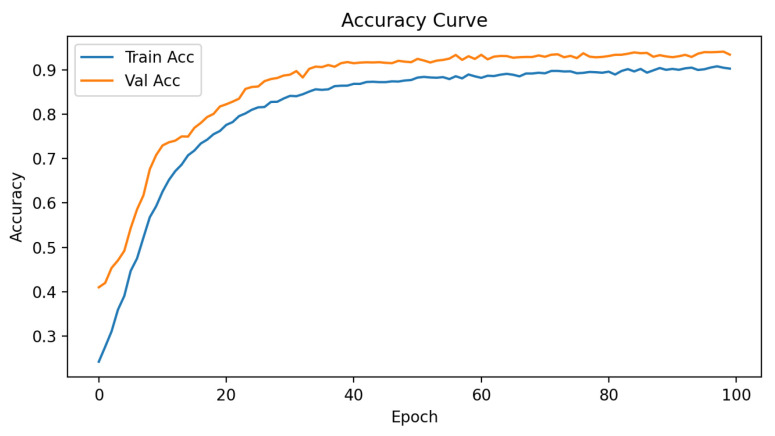
Training and validation trajectories (loss/accuracy) for the final classification MLP.

**Figure 7 sensors-25-07471-f007:**
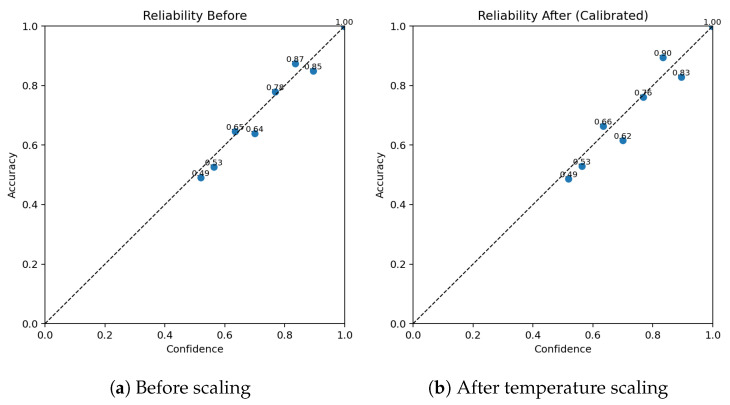
Classifier probability calibration reliability diagrams pre- and post-temperature scaling.

**Figure 8 sensors-25-07471-f008:**
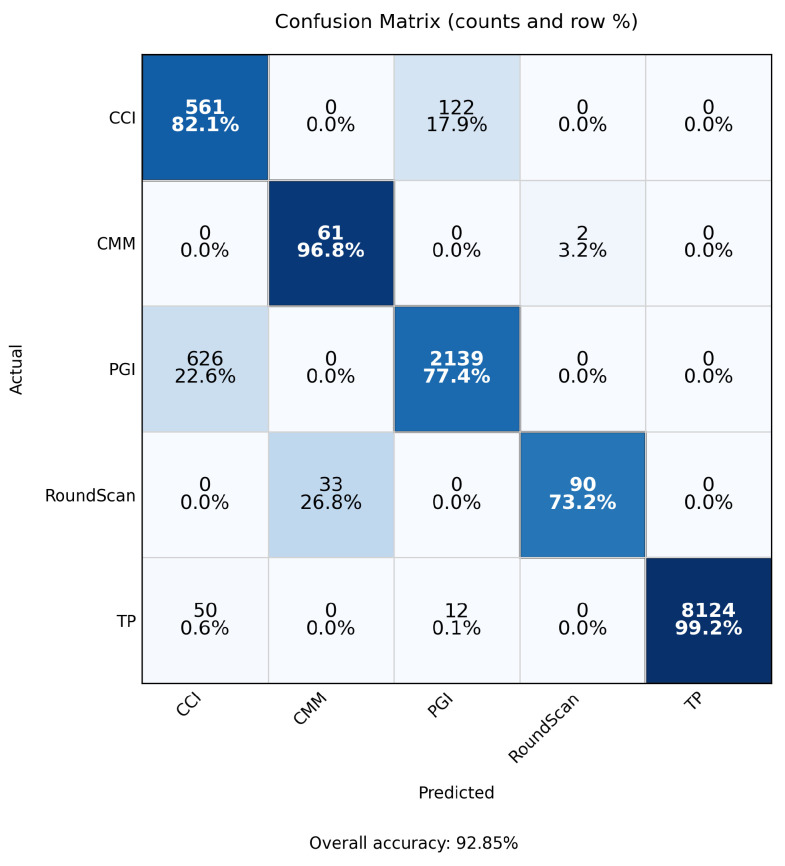
Confusion matrix of the calibrated classification model (system type prediction).

**Figure 12 sensors-25-07471-f012:**
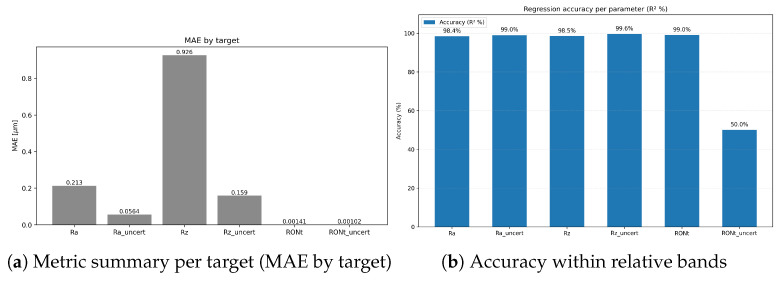
Single-target regression performance: aggregate metrics and tolerance-based accuracies for Ra, Rz, RONt.

**Table 1 sensors-25-07471-t001:** Example rows illustrating the five measurement system types used in the data collection. Binary flags: standard (reference specimen indicator), F (filter applied).

System_Type	Ra [μm]	Ra_uncert [μm]	Rz [μm]	Rz_uncert [μm]	Material	RONt [μm]	RONt_uncert [μm]	Standard	F	filtr_lc [mm]	filtr_ls [mm]	odc_el_lr [mm]
TP	0.83	0.09	3.15	0	1	0	0	1	1	0.8	0	0.80
PGI	0.07	0	1.71	0	1	0	0	0	0	0	0	0.75
CCI	0	0	0.34	0	1	0	0	0	0	0	0	0
CMM	0	0	0	0	6	0.39	0.01	1	1	0	0	0
RoundScan	0	0	0	0	1	1.43	0.21	1	0	0	0	0

Rows follow the same schema; magnitudes span orders between roughness and roundness parameters, and instruments include tactile (TP), optical (PGI/CCI) and form (CMM/RoundScan) systems.

**Table 2 sensors-25-07471-t002:** Regression performance metrics: single-target vs. weighted multi-output vs. classical baseline.

Target	Single-Target MLP			Multi-Output MLP (Weighted)			Baseline (HistGBM)		
	MAE [μm]	RMSE [μm]	*R* ^2^	MAE [μm]	RMSE [μm]	*R* ^2^	MAE [μm]	RMSE [μm]	*R* ^2^
Ra	0.2134	0.3730	0.9824	0.8695	1.7070	0.6323	0.1219	0.2287	0.9934
Rz	0.9255	1.5567	0.9847	4.2072	8.1861	0.5757	0.4852	0.8800	0.9950
RONt	0.00141	0.01339	0.9918	0.00124	0.01232	0.9930	0.00154	0.01494	0.9885
Ra_uncert	0.05639	0.08389	0.9899	0.2699	0.7708	0.1428	–	–	–
Rz_uncert	0.1589	0.3578	0.9955	1.5412	4.8790	0.1550	–	–	–
RONt_uncert	0.001020	0.01039	0.4934	0.001094	0.01208	0.3151	–	–	–
Mean (single-target)	0.2261	0.3990	0.9063						
Mean (multi-output)				1.1484	2.4329	0.4689			

Uncertainty target names retain the _uncert suffix. MAE and RMSE are reported in [μm]. Values rounded to three decimal places (four for $R^{2}$). Classical baseline results (Histogram Gradient Boosting Regressor) are shown only for the three primary parameters (Ra, Rz, RONt); uncertainty targets were not modelled with tree-based ensembles. Single-target MLPs generally provide higher fidelity for primary parameters and often their uncertainties compared to the weighted multi-output trunk. For the primary parameters, their MAE/RMSE and $R^{2}$ are competitive with the strong Histogram Gradient Boosting baseline, while additionally supporting direct modelling of uncertainty targets and integration with distributional heads.

**Table 3 sensors-25-07471-t003:** Multi-output loss variant comparison (averages across six targets).

Variant	Mean MAE [μm]	Mean $R^{2}$	Notes
Baseline (final)	1.325	0.582	Log-Huber; best mean $R^{2}$ but higher MAE
MAE	1.143	0.502	Lower MAE; weaker variance capture
Weighted MAE	1.148	0.469	Emphasises Ra, Rz; preserves RONt MAE
Log-Huber (alt)	1.325	0.582	Robust to outliers; similar to baseline

Values rounded to three decimals; metrics obtained from held-out validation summaries.

**Table 4 sensors-25-07471-t004:** Empirical coverage (EC) vs. nominal coverage (NC) for central prediction intervals before (quantile) and after conformal adjustment.

Target	Nominal	Quant EC	|Δ|	Conf EC	|Δ|	Quant W	Conf W
Ra	0.9	0.983	0.083	0.905	0.005	1.212	0.67
Rz	0.9	0.302	0.598	0.901	0.001	3.675	3.052
RONt	0.9	0.151	0.749	0.899	0.001	0.047	0

Quantile empirical coverage values (EC) and nominal coverage (NC) are shown as fractions (0–1); interval widths (W) are reported in [μm]. Conformal coverage reflects post-adjustment performance. Uncertainty target names retain the *_uncert suffix.

**Table 5 sensors-25-07471-t005:** Per-class precision, recall, and F1-scores for the calibrated classification model (support denotes number of evaluation samples per class). Overall accuracy: 92.85%.

Class	Precision	Recall	F1-Score	Support
CCI	0.454	0.821	0.584	683
CMM	0.649	0.968	0.777	63
PGI	0.941	0.774	0.849	2765
RoundScan	0.978	0.732	0.837	123
TP	1.000	0.992	0.996	8186
Macro avg	0.804	0.857	0.809	11,820
Weighted avg	0.953	0.929	0.935	11,820

Rounded to three decimal places.

## Data Availability

All code, processing scripts, trained-model artefacts (regeneration scripts), are available under the MIT License at the project repository (GitHub, latest commit snapshot) and archived on Zenodo at DOI: [31]. The release bundle includes hash manifests, ensuring integrity verification.

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
