# Peer review of "Multi-Task Deep Learning for Surface Metrology"

_sensors, 2025, doi:10.3390/s25247471_

Round 1

Reviewer 1 Report

Comments and Suggestions for Authors

This article presents a deep learning framework for predicting surface texture parameters (Ra, Rz, RONt) and their associated standard uncertainties using a multi-instrument dataset. It explores classification of measurement system types and uncertainty quantification using quantile regression, heteroscedastic regression, and conformal prediction. Single-target models outperform multi-output ones due to heterogeneous noise scales. The framework is publicly available and demonstrates practical utility for informed metrological decision-making. However, here are some comments that can help improve the paper:

  • Methodology: The rationale behind choosing specific quantile levels or architecture parameters (like dropout rates) could be better justified or benchmarked against alternatives.

  • Comparison: Include a quantitative comparison with traditional models or other recent deep learning techniques on the same or similar datasets. Consider benchmarks like random forests or gradient boosting for a more rigorous comparison.

  • Results: The performance metrics are comprehensive, but the result reporting is overly focused on R² without sufficient discussion of the implications of interval widths or tolerance-based metrics in practical terms. More explanation is needed on the interpretation of the lower R² for RONt_uncert and how that impacts real-world decisions.
  • Future Work: While the discussion outlines several future directions, including local conformal scaling and active learning, these are quite generic. Further expand some possible experimental designs to explore that you think can further improve your work.

Author Response

The authors thank the reviewer for their valuable comments, which helped improve the paper. We agree with the comments and have prepared a revised version of the manuscript. All corresponding changes have been highlighted in colour in the attached marked version of the manuscript.

Comments 1: Methodology: The rationale behind choosing specific quantile levels or architecture parameters (like dropout rates) could be better justified or benchmarked against alternatives.

Response 1:

We thank the reviewer for this constructive methodological suggestion. In the revised manuscript, we have clarified the rationale for both the chosen quantile levels and the network architecture/hyperparameters.

For quantile regression, we now explain that the target quantiles (q \in {0.05, 0.10, 0.50, 0.90, 0.95}) were selected to approximate the central coverage ranges (90–95 %) commonly used in metrology when reporting measurement uncertainty. We also explicitly distinguish a narrower inner band (10–90 %) and a wider outer band (5–95 %) to support sensitivity analyses. This clarification is provided in Section 3.3 (“Quantile regression”), where we link these levels to metrological conventions and to the interpretation of interval widths.

For the network architecture and hyperparameters (depth, width, dropout, L2 regularisation), we now state that they were selected via a coarse grid search over depths 3–5 and widths 64–512 on the validation split, rather than being chosen ad hoc. We emphasise that the final architectures reflect a trade‑off between accuracy, stability and simplicity. This is described in Section 3.2 (“Baseline deterministic models”), in the paragraph titled “Architecture selection”.

Comment 2:
Comparison: Include a quantitative comparison with traditional models or other recent deep learning techniques on the same or similar datasets. Consider benchmarks like random forests or gradient boosting for a more rigorous comparison.
Response 2:

We agree that an explicit quantitative comparison with strong traditional baselines strengthens the contribution. In the revised manuscript, we now more clearly document and summarise the classical models trained alongside our deep architectures.

In Section 3.2 (“Baseline deterministic models”), we have added and expanded the paragraph “Classical baselines”, explicitly stating that classical tabular methods (random forests, gradient boosting and k‑nearest neighbours) were trained on the same features and targets as the deep models. This paragraph now includes a concise quantitative summary of the strongest baseline: on the held‑out split, a tuned Histogram Gradient Boosting Regressor achieves high accuracy for all three primary parameters (e.g. for (Ra): MAE (\approx 0.12), RMSE (\approx 0.23), (R^2 \approx 0.993); with similarly high (R^2) for (Rz) and (RONt)). We clarify that these results are broadly comparable to those of the single‑target MLP regressors, with small differences in MAE/RMSE and (R^2) across targets, while the deep models additionally support direct prediction of the uncertainty targets and integration with quantile, heteroscedastic and conformal components within a unified classifier–regressor pipeline. In other words, the tree‑based ensembles serve as strong state‑of‑the‑art point‑prediction baselines, whereas the deep framework provides an integrated classifier–regressor pipeline with calibrated, model‑based uncertainty.

We have also updated the main regression summary table (Table 2 in Section 4) to explicitly include these classical baselines. The table now reports MAE, RMSE and (R^2) side‑by‑side for three model families: (i) single‑target MLPs, (ii) weighted multi‑output MLPs, and (iii) the Histogram Gradient Boosting baseline, for the three primary parameters (Ra), (Rz) and (RONt). For the uncertainty targets ((\texttt{Ra_uncert}), (\texttt{Rz_uncert}), (\texttt{RONt_uncert})), the baseline columns are marked “--”, and the caption explains that these were not modelled with tree‑based ensembles.

Together, these changes provide the requested quantitative comparison against widely used classical baselines on the same dataset and clearly position our deep models as competitive in terms of point accuracy, while offering an integrated classifier–regressor pipeline with richer uncertainty and distributional outputs. All corresponding modifications are highlighted in colour in the marked version of the revised manuscript.

Comment 3: Results: The performance metrics are comprehensive, but the result reporting is overly focused on R² without sufficient discussion of the implications of interval widths or tolerance-based metrics in practical terms. More explanation is needed on the interpretation of the lower R² for RONt_uncert and how that impacts real-world decisions.
Response 3:

We appreciate this important point and have strengthened the practical interpretation of our results.

First, we now explicitly emphasise the role of tolerance‑style metrics (relative and absolute bands) as central to practice, because they directly reflect the decision thresholds used by practitioners (e.g. acceptance within a specified percentage or absolute tolerance). In the Discussion (Section 5), the paragraph beginning “Three practical observations emerge” has been updated to state that tolerance metrics complement MAE/RMSE and are critical when assessing whether predictive performance is sufficient for conformity and process decisions.

Second, we clarify how to interpret the interval widths from the quantile and heteroscedastic models (with conformal adjustment). Reported intervals are expressed in micrometres [(\mu)m]. For (Ra) and (Rz), the interval magnitudes are broadly comparable to empirically reported standard uncertainties, so they can meaningfully support or complement experimental evaluation when repeated acquisitions are impractical. This is discussed in Section 5 (“Uncertainty quantification benefited from a layered design…”) and in the coverage/width explanation linked to the interval‑coverage table.

Third, we expanded the discussion of (\texttt{RONt_uncert}) in a dedicated “RONt‑specific considerations” paragraph in Section 5. There, we explain that the lower (R^2) for (\texttt{RONt_uncert}) is mainly due to (i) instrument heterogeneity across different roundness testers and setups, which introduces domain shift, and (ii) partially specified uncertainty budgets and higher label noise for roundness, which inherently limit the attainable (R^2). We explicitly note that (\texttt{RONt_uncert}) predictions should therefore be interpreted with greater caution, and that well‑calibrated coverage and tolerance‑based metrics are more informative than (R^2) alone in this case. We also outline possible mitigations (harmonised acquisition protocols, richer instrument metadata, conditional heads, more complete uncertainty budgets).

These revisions place interval widths and tolerance metrics at the centre of the practical interpretation and clarify how the lower (R^2) for (\texttt{RONt_uncert}) affects real‑world use.

Comment 4:
Future Work: While the discussion outlines several future directions, including local conformal scaling and active learning, these are quite generic. Further expand on possible experimental designs you think could improve your work.

Response 4:

We agree and have made the future‑work section more concrete.

In the Outlook paragraph of the Discussion (Section 5), we now describe a more specific implementation for local conformal scaling. This uses non‑conformity scores normalised with local or conditional scale estimates—e.g. based on nearest‑neighbour or heteroscedastic predictions. The resulting intervals are then evaluated in underrepresented regions of the feature space using coverage and width criteria.

For active learning, we outline an experiment in which the current model is used to select new measurements (material–system–parameter combinations) from regions with high predictive uncertainty or poor calibration. New data from these regions is then acquired and the model retrained, tracking improvements in coverage and tolerance‑based accuracies after each acquisition round.

We also retain additional concrete directions, such as adaptive loss reweighting and instrument‑conditioned heads, as listed in the bullet‑style “Outlook” paragraph.

These additions give readers clear starting points for follow‑up studies that extend the present framework.

Reviewer 2 Report

Comments and Suggestions for Authors

General comment:

This manuscript introduces a reproducible learning framework for surface metrology applications. The goal is to predict surface texture parameters along with their standard uncertainties. The proposal includes a classifier and a regression stage exhibiting acceptable accuracy. The main contribution is the application of machine learning methods in surface metrology. Some comments should be addressed before the manuscript can be accepted for publication.

Comment 1:

In the abstract, some “variables” and “acronyms” are not easy to follow for the readership. It is desirable to avoid using such names to improve the paper's readability.

Comment 2:

Up to lines 157-158, it is not easy to follow the meaning of Ra, Rz, and RONt. I recommend defining them earlier or providing a table with the abbreviations and variables.

Comment 3:

Besides the excellent explanation of the learning architecture, I always expect to read some scientific rationale on why the model/algorithm was chosen. That is, providing details on the architecture is nice, but the readership should have a formal background on how it works and why it works for your specific problem. For instance, what justifies (formally) the use of an NLL mode?

Comment 4:

Perhaps obvious, but you should provide the formulae for the MAE, RMSE, …

Comment 5:

A comparison with SOTA methods would strengthen the paper's contribution.

Author Response

The authors thank the reviewer for their valuable comments, which helped improve the paper. We agree with the comments and have prepared a revised version of the manuscript. All corresponding changes have been highlighted in colour in the attached marked version of the manuscript.

Comment 1:

In the abstract, some “variables” and “acronyms” are not easy to follow for the readership. It is desirable to avoid using such names to improve the paper's readability.

Response 1: 
We thank the Reviewer for this helpful suggestion. The abstract has been rewritten to reduce unexplained notation and to improve readability. In particular, we now explicitly name the main surface parameters as arithmetic mean roughness (Ra), mean peak‑to‑valley roughness (Rz) and total roundness deviation (RONt), and we refer to “standard uncertainties” in plain language instead of coded variables such as *\_uncert. We also describe the main performance indicators in words (e.g. “coefficients of determination” and “expected calibration error”) rather than relying on bare symbols. These changes appear in the revised Abstract.

Comment 2:

Up to lines 157-158, it is not easy to follow the meaning of Ra, Rz, and RONt. I recommend defining them earlier or providing a table with the abbreviations and variables.

Response 2:

We agree that early definitions improve accessibility. In the Introduction, we have added a short paragraph that explicitly defines the three main surface parameters: Ra, the arithmetic mean roughness; Rz, the mean peak‑to‑valley roughness; and RONt, the total roundness deviation. The same paragraph states that these parameters and their associated standard uncertainties are the primary targets for prediction and the primary basis for judging metrological adequacy in this study. This addition appears in the Introduction.

Comment 3:

Besides the excellent explanation of the learning architecture, I always expect to read some scientific rationale on why the model/algorithm was chosen. That is, providing details on the architecture is nice, but the readership should have a formal background on how it works and why it works for your specific problem. For instance, what justifies (formally) the use of an NLL mode?

We appreciate this request for clarification. First, in the Methods section we now explicitly motivate the choice of multilayer perceptrons as the backbone architecture: our predictors are mixed tabular descriptors (materials, flags, filter settings), for which fully connected networks are a natural choice; under mild regularity assumptions, they can approximate generic nonlinear mappings, and they have been widely validated on related metrology and manufacturing problems. Second, in the Heteroscedastic Gaussian regression subsection, we have added a short explanation that minimising the Gaussian negative log‑likelihood corresponds to maximum likelihood estimation under a heteroscedastic Gaussian noise model and constitutes a standard proper scoring rule for jointly learning the conditional mean and variance. This clarification appears in Section 3 (Method).

Comment 4:

Perhaps obvious, but you should provide the formulae for the MAE, RMSE, …

Response 4:

We agree and have now added explicit formulas in the Evaluation metrics subsection. These equations now appear directly in the Evaluation metrics subsection of the Methods (Section 3).

Comment 5:

A comparison with SOTA methods would strengthen the paper's contribution.

Response 5: 

We appreciate this suggestion and have strengthened the connection to state‑of‑the‑art methods for tabular regression. In our setting, strong tree‑based ensembles such as random forests and gradient boosting are widely regarded as state‑of‑the‑art baselines for structured tabular data and are frequently used as references in recent surface‑metrology and manufacturing studies. Accordingly, we trained several such models (including Gradient Boosting and Histogram Gradient Boosting) on exactly the same features and targets as our deep models.

In the revised manuscript, we now (i) explicitly describe these classical baselines in the “Classical baselines” paragraph of Section 3.2 (“Baseline deterministic models”), and (ii) report their quantitative performance alongside our neural networks in the main regression summary table (Table 2). For the three primary parameters ((Ra), (Rz) and (RONt)), the best classical baseline (Histogram Gradient Boosting Regressor) achieves very high (R^2) values (around 0.99) with low MAE and RMSE, and the single‑target MLP regressors attain broadly comparable accuracy, with differences in MAE/RMSE and (R^2) remaining within the same order of magnitude. The main advantage of the deep models is that the same backbone architecture extends naturally to distributional heads (quantile and heteroscedastic) and supports conformal post‑calibration, enabling calibrated prediction intervals, tolerance‑style metrics and direct prediction of the reported standard uncertainties. In addition, the classifier and regressors operate on the same tabular descriptors and share design principles, forming an integrated classifier–regressor pipeline with calibrated, model‑based uncertainty; classical tree‑based baselines, while strong for point prediction, do not natively provide this joint classification–regression framework with uncertainty quantification.

These additions clarify that our deep learning framework is competitive with strong state‑of‑the‑art tree‑based baselines on this dataset, while offering an integrated classifier–regressor pipeline and richer metrological information for decision support. All related changes are highlighted in colour in the marked version of the revised manuscript.

Reviewer 3 Report

Comments and Suggestions for Authors

I think the working direction is excellent. However, before I can make my final decision, I humbly requested following from the authors:

1, Since there are so many co-authors, could the writing be improved? I have difficulty in understanding a lot of things in the paper, from the very beginning.

2, The scientific goal and why it is significant are not well address. I hope to see the reasoning.

3, For example, predict measurement system type is the first research problem, but I do not understand why it is important, and I did not find simple explanations.

Author Response

The authors thank the reviewer for their valuable comments, which helped improve the paper. We agree with the comments and have prepared a revised version of the manuscript. All corresponding changes have been highlighted in colour in the attached marked version of the manuscript.

Comment 1:
Since there are so many co-authors, could the writing be improved? I have difficulty in understanding a lot of things in the paper, from the very beginning. Response 1:
Thank you for your feedback. We revised the manuscript to improve clarity, especially in the early sections.

  • In Section 1 (Introduction), we rewrote the objective and motivation paragraphs to state in simple terms what is being predicted (surface parameters and their standard uncertainties), on what data (multi-instrument measurements), and for what purpose (instrument selection, conformity assessment, and process monitoring).
  • We also ensured that key terms (surface parameters, standard uncertainties, measurement system types, uncertainty intervals) were introduced and defined straightforwardly on first use.
  • The opening of Section 2 (Method) now directly connects to the clarified objective and summarises the overall pipeline before presenting the technical details.

Comment 2:
The scientific goal and its significance are not well addressed. I hope to see the reasoning.

Response 2:
We agree and have clarified the scientific goal and its importance.

  • In Section 1 (Introduction), we have added a clear statement of the primary objective: to develop and rigorously evaluate a data‑driven framework that simultaneously predicts (Ra), (Rz), and (RONt), and their standard uncertainties, across multiple measurement systems.
  • We now explain why this matters for metrology: the framework provides fast, consistent, traceable predictions that support key decisions, such as instrument selection, conformity assessment, and process monitoring, where both the nominal value and uncertainty matter.
  • We clarify that our contribution complements existing metrology practice by providing uncertainty-aware predictions and does not replace formal uncertainty evaluation when required by standards.

We made these additions to the new objective/motivation paragraphs at the end of Section 1 and reflected them in the abstract and conclusions.

Comment 3:
For example, predicting the measurement system type is the first research problem, but I do not understand why it is essential, and I did not find simple explanations.

Response 3:
Thank you for pointing this out. We have now given a straightforward explanation of why predicting measurement system type matters.

  • In Section 1 (Introduction), we explain that each measurement system (tactile, optical, or form) has a typical noise and uncertainty profile. Predicting system type from data is helpful when metadata are incomplete or inconsistent, and it identifies the relevant noise regime.
  • We emphasise that the system‑type classifier is not only a separate prediction task but also an integral part of the framework: it can flag suspicious measurements when the predicted system type differs from the recorded label, and it provides a natural conditioning signal for the subsequent regression and uncertainty models.
  • In Section 3 (Results), after presenting the classification performance, we give a short, direct explanation: the classifier acts as a sanity check, metadata aid, and input for system-aware modelling, not as an end goal.

We hope these changes clarify the purpose and importance of system-type prediction.

According to the reviewer's suggestion, the English have been improved in the manuscript (minor revision).

Round 2

Reviewer 3 Report

Comments and Suggestions for Authors

Since this is a framework, as you said, then you should provide an illustration, together with input and output. 

For type prediction, my doubt is, given the data, we should have already known what type of system is used. I cannot imagine that the type is unknown.

Such things are where I am lost.

Either you tell me what is the knowledge that I am supposed to know but unfortunately I do not know, or please make your presentation clear.

Author Response

Please find attached the marked changes based on your comments. Here are the responses:

Comment 1 :
"Since this is a framework, as you said, then you should provide an illustration, together with input and output."

Response 1:
We thank the Reviewer for this suggestion. In the revised manuscript we have added a dedicated schematic of the full pipeline (new Fig. 3). This figure shows, on a single page, the tabular inputs (system type label, material, reference/standard flag, filter and cut‑off settings, evaluation length, etc.), the shared tabular MLP backbone, the upper branch for measurement system classification (tactile/optical/form), and the lower branch of six single‑target regressors for (Ra), (Rz), (RONt) and their reported standard uncertainties, together with the quantile, heteroscedastic and conformal components used to obtain calibrated prediction intervals. The outputs (calibrated point predictions in (μm), prediction intervals, and tolerance‑style metrics) are explicitly indicated. In the Method overview, we also added a linking sentence: “The overall pipeline, including the shared backbone, classifier branch and single‑target regression with uncertainty stack, is summarised schematically in Fig.3.”

Comment 2 :
"For type prediction, my doubt is, given the data, we should have already known what type of system is used. I cannot imagine that the type is unknown."

Response 2:
We agree that, in our dataset, the measurement system type is known and recorded. Learning is therefore performed in the standard supervised setting: models are trained on a subset of labelled records and evaluated on a held‑out subset of previously unseen data. To clarify why we nevertheless predict system type, we have expanded the “Problem formulation” section. The revised text explains that the system‑type predictor is used primarily as a metadata sanity check and input to system‑aware regression and uncertainty models in large multi‑instrument databases, where inconsistent or missing entries can occur (the classifier flags potential issues when the predicted type disagrees with the recorded label and indicates which noise regime a given sample most likely belongs to). In addition, we now make explicit that the same mechanism is intended as a building block for future decision‑support tools: in realistic scenarios, the material, geometry and process state are known, but the choice of a suitable measurement system among many available instruments is non‑trivial and often resolved by expert discussion. We therefore envisage our classifier and regressors as components of an intelligent, database‑driven “metrology assistant” that can recommend appropriate instrument families and operating conditions and integrate naturally with automated and robotic metrology cells. This clarifies that we are not trying to infer a fundamentally unknown label, but to encode and automate practically useful knowledge for measurement planning.

Comment 3 :
"Such things are where I am lost."

Response 3:
We apologise that the original presentation did not make these roles sufficiently transparent. To address this, we have:

  • added the new framework figure with explicit inputs, internal components and outputs, and
  • clarified in “Problem formulation” how the classifier and regressors are intended to be used in practice (metadata validation, instrument‑selection support, and uncertainty‑aware prediction of surface parameters). We hope this improves readability and makes it easier to follow the framework's logic.

Comment 4:
"Either you tell me what is the knowledge that I am supposed to know but unfortunately I do not know, or please make your presentation clear."

Response 4:
We appreciate this remark and have sought to make the underlying “knowledge” explicit rather than implicit. In the revised manuscript we now:

  • state clearly that system‑type learning is based on labelled data (known instrument types) with standard training/validation/test splits,
  • describe the practical knowledge encoded by the classifier (characteristic noise and uncertainty profiles of tactile, optical and form systems, used for metadata checking and future measurement planning), and
  • provide a visual summary (Fig. 3) that connects these components to their inputs and outputs. Together, these changes are intended to make the presentation clearer and to remove the need for any unstated prior knowledge on the reader’s side.
